# Release of Cholecystokinin from Rat Intestinal Mucosal Cells and the Enteroendocrine Cell Line STC-1 in Response to Maleic and Succinic Acid, Fermentation Products of Alcoholic Beverages

**DOI:** 10.3390/ijms21020589

**Published:** 2020-01-16

**Authors:** Jan-Hendrik Egberts, Ghulam Shere Raza, Cornelia Wilgus, Stefan Teyssen, Karlheinz Kiehne, Karl-Heinz Herzig

**Affiliations:** 1University Hospital Schleswig-Holstein, Campus Kiel, 24105 Kiel, Germany; Jan-Hendrik.Egberts@uksh.de (J.-H.E.); cwilgus@dermatology.uni-kiel.de (C.W.); Karlheinz.Kiehne@web.de (K.K.); 2Research Unit of Biomedicine, University of Oulu, 90014 Oulu, Finland; ghulam.raza@oulu.fi; 3Krankenhaus St. Joseph-Stift, 28209 Bremen, Germany; stephan.teyssen@hirslanden.ch; 4Department of Gastroenterology and Metabolism, Poznan University of Medical Sciences, 61-701 Poznan, Poland; 5Medical Research Center Oulu and Oulu University Hospital, 90014 Oulu, Finland

**Keywords:** cholecystokinin (CCK), enteroendocrine cells STC-1, pancreatic secretions, pancreatitis, alcohol, fermentation, maleic and succinic acids, intracellular calcium

## Abstract

Alcoholic beverages stimulate pancreatic enzyme secretions by inducing cholecystokinin (CCK) release. CCK is the major stimulatory hormone of pancreatic exocrine secretions, secreted from enteroendocrine I-cells of the intestine. Fermentation products of alcoholic beverages, such as maleic and succinic acids, influence gastric acid secretions. We hypothesize that maleic and succinic acids stimulate pancreatic exocrine secretions during beer and wine ingestion by increasing CCK secretions. Therefore, the effects of maleic and succinic acids on CCK release were studied in duodenal mucosal cells and the enteroendocrine cell line STC-1. Mucosal cells were perfused for 30 min with 5 min sampling intervals, STC-1 cells were studied under static incubation for 15 min, and supernatants were collected for CCK measurements. Succinate and maleate-induced CCK release were investigated. Succinate and maleate doses dependently stimulated CCK secretions from mucosal cells and STC-1 cells. Diltiazem, a calcium channel blocker, significantly inhibited succinate and maleate-induced CCK secretions from mucosal cells and STC-1 cells. Maleate and succinate did not show cytotoxicity in STC-1 cells. Our results indicate that succinate and maleate are novel CCK-releasing factors in fermented alcoholic beverages and could contribute to pancreatic exocrine secretions and their pathophysiology.

## 1. Introduction

Cholecystokinin (CCK) is the major stimulatory hormone of pancreatic exocrine secretions, secreted in response to intraluminal nutrients by the enteroendocrine I-cells of the duodenum [1]. Various substances, like peptides, essential amino acids, and fatty acids, stimulate CCK release via calcium-sensing receptors and GPR40, respectively [2,3]. Physiological doses of CCK-8 induce pancreatic enzyme secretions, and supramaximal doses of CCK-8 cause inflammation and cell deaths, which are features of human pancreatitis [4]. Pancreatitis is an inflammatory condition that leads to acinar atrophy and fibrosis. A major etiological factor is alcohol abuse [5,6]. About one-third of acute pancreatitis cases in the United States are alcohol-induced, and 60%–90% of pancreatitis patients have a history of chronic alcohol consumption [7]. It is estimated that drinking more than 80 gm of alcohol/day for 6–12 years is required to cause pancreatitis [8]. A variety of alcoholic beverages stimulate pancreatic enzyme secretions, contributing to the development of pancreatitis [9,10]. Pure alcohol itself is a rather weak stimulator of pancreatic secretions and some studies even reported an inhibition [11]. A population-based study in Denmark showed a two-fold risk of pancreatitis in a cohort of 17,905 men and women drinking more than 14 beers/week, compared to wine and spirits [12]. It has been demonstrated that alcohol and alcoholic beverages that increase plasma levels of ethanol do not affect pancreatic enzyme secretions in humans [13]. In contrast, it has been shown that beer in lower doses (250 mL) increased pancreatic enzymes (trypsin 1053% and amylase 323%), while in higher doses (850 mL), it did not [9]. The authors suggested that circulating ethanol neutralized the stimulatory effect of the nonalcoholic components of beer on pancreatic secretions [9].

However, alcoholic beverages contain numerous nonalcoholic compounds that could have beneficial or harmful effects on pancreatic functions [14]. About 2000 organic and inorganic constituents were identified in the beer and 1000 in wine [14]. Pancreatic secretion studies in humans suggested that nonalcoholic constituents of alcoholic beverages might be responsible for pancreatic enzyme secretions [15]. An in vitro study using freshly isolated pancreatic acinar cells and the AR4-2J cell lines demonstrated that beer stimulates amylase secretions at the same magnitude as the maximally effective concentrations of CCK [16]. Teyssen et al. demonstrated that alcoholic beverages produced by fermentation stimulate gastric acid output and gastrin release in healthy humans [17]. The same group found that fermentation products’ maleic and succinic acids were powerful stimulants of gastric acid output in humans [18]. Beer contains approximately 135 µM succinic acid and 948 µM maleic acid [18].

Under physiological conditions, plasma succinate concentrations in rodents range from 6 to 20 μM and, in humans, 2–20 μM, whereas lower levels were found in the serum [19,20]. Succinate is a ligand for orphan G-protein coupled receptor GPR91 and acts as a signaling molecule. Maleate also activates GPR91 but is 5–10-fold less potent than succinate [21]. The authors reported that half-maximal response concentrations of succinate were 56 ± 8 µM in humans and 28 ± 5 µM in mice, indicating that under physiological conditions the levels are too low to activate GPR91 [21].

We hypothesize that maleic and succinic acids stimulate pancreatic secretions via release of CCK after ingestion of alcoholic beverages. In the present study, the effects of maleate and succinate on CCK release from isolated duodenal mucosal cells and enteroendocrine STC-1 cells were investigated.

## 2. Results

### 2.1. Succinate and Maleate Stimulate CCK Secretions from Mucosal Cells and STC-1 Cells

The basal CCK secretions from intestinal mucosa cells were 3.9 pmol/30 min. Succinate dose dependently stimulated CCK release to a maximum of 23.3 pmol at 10^−4^ M succinate (Figure 1A). The half-maximal response was obtained with 10^−6^ M succinate. Maleate caused a similar dose response curve from duodenal mucosal cells (Figure 1B), inducing a maximum secretion of 13.1 pmol CCK by 10^−3^ M maleate and the half-maximal stimulation by 10^−5^ M maleate.

We further studied succinate (Figure 2A) and maleate-stimulated (Figure 2B) CCK secretions using the enteroendocrine cell line STC-1. Basal CCK release from STC-1 cells was 4.1 pmol/15 min. Succinate (10^−5^ M) caused a maximum CCK release of 10.5 pmol and a half-maximal response at 10^−7^ M succinate. Maleate (10^−5^ M) caused a maximum CCK release of 9.1 pmol, with a half-maximal response at 10^−7^ M maleate from STC-1 cells.

Additional factors released from the intestinal mucosa and STC-1 cells may also influence amylase release from pancreatic acini. A specific CCK1 receptor antagonist (L-364718) completely inhibited amylase release from pancreatic acini in the bioassay from the eluates from intestinal mucosal cells and the supernatant from STC-1 cells; therefore, CCK values did not differ from the basal. In addition, no direct effect on amylase secretion was observed with maleate or succinate on freshly isolated pancreatic acini.

### 2.2. Calcium Channel Inhibitor: Diltiazem Inhibits Succinate and Maleate-Stimulated CCK Releases

Diltiazem, an L-type voltage-sensitive calcium channel (VSCC) inhibitor, might affect amylase release from pancreatic acini in the bioassay. Therefore, a radioimmunoassay (RIA) kit was used for measurement of CCK from STC-1 cells. Preincubation of STC-1 cells (Figure 3) with 10^−5^ M diltiazem (calcium channel inhibitor) reduced 2.5 mM succinate-stimulated CCK secretions, from 12.5 pmol CCK to 7.1 pmol (43% inhibition). Maleate-stimulated CCK secretions (5 mM) were also reduced by 10^−5^ M diltiazem, from 16.5 pmol to 10.5 pmol CCK (36.4% inhibition). The intra-assay CV was <14% and interassay CV was with 10% within the limits provided by the manufacturer.

### 2.3. Lactate Dehydrogenase (LDH) Release with Succinate and Maleate for Cytotoxicity

Both succinate and maleate significantly inhibited lactate dehydrogenase (LDH) release, compared to the maximum LDH release control from STC-1 cells (Figure 4). The LDH releases with 5 mM succinate and 5 mM maleate were 2.4% and 2.6%, respectively, compared to a maximum LDH release (100%). A buffer caused a 2.2% LDH release.

## 3. Discussion

Our results demonstrate that the fermentation products succinate and maleate stimulate CCK release from duodenal mucosal cells and the enteroendocrine cell line STC-1. l-type calcium channels play a central part in the release of CCK from intestinal mucosa [22,23,24]. The increase in intracellular calcium concentrations are regulated by receptor-dependent activations of calcium channels on the extracellular membrane and intracellular calcium stores [25]. We found that l-type calcium channel blocker diltiazem inhibited the succinate and maleate-induced CCK releases from the enteroendocrine cells. Our results indicate that l-type calcium channel blocker diltiazem inhibits CCK secretions from STC-1 cells.

The duodenum is a central site for the regulation of pancreatic secretions. The specialized endocrine cells in the mucosa respond to the ingested nutrients with a release of gastrointestinal hormones like CCK into the blood, which leads to increased pancreatic secretions [26]. CCK is synthesized in endocrine I-cells, pituitary corticotrophs and melanotrophs, in thyroid C-cells, and adrenal medullary cells [27,28]. In circulation, CCK originates mainly from intestinal endocrine cells. CCK mediates its effects through two receptors: CCK1 and CCK2, which are G-protein-coupled receptors. CCK1 is expressed in gallbladder, gastric mucosa, pancreatic acinar cells, and certain areas of the central and peripheral nervous systems, while CCK2 is expressed in the stomach, human pancreas, and central nervous system (CNS) [29]. The CCK1 receptor is highly specific and binds O-sulfated CCK peptides with high affinity, while CCK2 receptor is less specific and binds both sulfated and nonsulfated CCK peptides and gastrin [30,31]. It has been shown that the CCK1 receptor is present with distinct cellular distributions in the exocrine pancreas of rodents, guinea pigs, and humans [32,33,34]. CCK stimulates exocrine secretions from the pancreas by activating a phospholipase C (PLC)-inositol triphosphate receptor (IP3R) signaling cascade that induces intracellular calcium releases from the endoplasmic reticulum [35].

Various food ingredients, such as fat, proteins, and amino acids, are the most potent stimulants of CCK secretions in humans [36]. Removal of extracellular calcium and/or application of the l-type voltage-gated calcium channel (VGCC) blocker nicardipine significantly reduced, e.g., the free fatty acids induced by intracellular calcium rises and CCK secretions in STC-1 cells [37,38]. Furthermore, bitter substances, such as denatonium benzoate, phenylthiocarbamide, and quinine, induce CCK secretions [39,40]. Yamazaki et al. demonstrated that mature hop bitter acid (MHBA) induced CCK secretions in STC-1 cells through increases in intracellular Ca^2+^ via l-type voltage-sensitive Ca^2+^ channels [41].

These compounds stimulate CCK secretions directly [36] and/or indirectly by CCK-releasing hormones from duodenal mucosa [26]. Animal and human studies on pancreatic secretions suggest that physiological doses of CCK activate neural pathways and the cholinergic receptors on the acinar cells [42]. Furthermore, cholinergic hyperstimulation of the pancreas has been implicated as a mechanism of alcohol-induced pancreatitis, including NF-κB (nuclear factor-kappa B) activation [43]. Ethanol itself might cause pancreatic injuries, e.g., by reducing pancreatic blood flow and microcirculation in stimulated organs, as shown in animal models [44]. In addition, animal studies have demonstrated that NF-κB activation is an early event in hormone-induced pancreatitis [4]. Experiments using rat pancreatic acini have shown that supraphysiological concentrations of CCK induces NF-κB through PKC (protein kinase C) activation and by increasing intracellular calcium [45]. These studies demonstrate that CCK and cholinergic stimulations of pancreatic acini induce pancreatitis. Animal studies have shown that CCK and its receptors are initiating factors in acute pancreatitis [46,47]. Pancreatic hyperstimulation with CCK is one potential factor in the pathophysiology of alcohol-induced pancreatitis [48].

In humans, it has been demonstrated that fermented alcoholic beverages stimulate gastric acid secretions [18]. Similar to the gastric acid secretions, pancreatic secretions are stimulated by fermented alcoholic beverages, while distilled alcoholic beverages do not stimulate the exocrine pancreas [9,10]. Succinate and maleate have been identified as stimulators of gastric secretions in response to the ingestion of beer or wine [18]. Succinate and maleate are produced by fermentation processes of alcoholic beverages and by yeast when glucose is fermented [9]. This might be the reason that fermented alcoholic beverage products like maleate and succinate induce CCK release, which further could contribute to the development of pancreatitis.

In summary, we have identified succinate and maleate as potential CCK-releasing factors in fermented alcoholic beverages acting on enteroendocrine cells, and thus, stimulating pancreatic exocrine secretions. Their contributions in the development of pancreatitis would need further investigations, e.g., by reducing their amounts in beer.

## 4. Materials and Methods

### 4.1. Materials

CCK-8 was purchased from Bachem, Bubendorf, Switzerland; L364718 from Merck, Darmstadt, Germany; collagenase (CLSPA grade) from Worthington, NJ, USA; amylase detection reagent from Boehringer Mannheim, Germany; EU-RIA CCK kit for CCK measurements from EuroDiagnostic, Malmö, Sweden; Sephadex G-50 from Pharmacia, Freiburg, Germany; and C18 Sep-Pac cartridges were from Water Corporation, Milford, MA, USA. All other reagents were purchased from Sigma, Deisenhofen, Germany.

### 4.2. Intestinal Mucosa Cell Perfusions

Mucosa cells were prepared from the proximal 20 cm small bowels of Wistar rats, as previously described [49,50]. The gut segments were opened longitudinally and washed with saline, followed by incubation in a calcium-free saline buffer containing 2.5 mM EDTA at 37 °C for 4 min. Detached cells were separated in a shaking water bath and collected by centrifugation at 300 *g* for 3 min. The pellets were resuspended in HEPES-buffered Ringer solution; filtered through gauze; mixed with swollen Sephadex G-50; and aliquots of 4 mL were transferred into perfusion columns (Renner KG, Dannstadt, Germany). Cells were perfused at 37 °C with a flow of 1 mL/min, and samples were collected for 5 min. The cells in the perfusion system were equilibrated with HEPES-buffered Ringer solution and perfused for 50 min prior to the experiment. Then, two fractions of 5 mL each were collected as a baseline. The cells in the perfusion columns were stimulated with increasing concentrations each of succinate and maleate, ranging from 10^−7^ M to 10^−3^ M, for 3 periods of 5 min. After the last stimulation period, further 5 min samples were collected. The samples were extracted on C18 Sep-Pac cartridges, eluted with acetonitrile–water (80:20), and eluates were dried under a flow of nitrogen and analyzed for CCK in bioassay.

### 4.3. Cell Cultures

The STC-1 cell line was derived from an intestinal endocrine tumor in a double-transgenic mouse [51]. These cells are widely used as a model enteroendocrine cell line for CCK [25] and GLP-1 release [3]. Stock cultures of STC-1 cells were maintained in Dulbecco’s modified Eagles medium (DMEM), supplemented with 2.5% fetal bovine serum and 15% horse serum in a humidified atmosphere containing 5% CO_2_ and 95% air at 37 °C. Cells were passaged after 4 days of growth. For the experiment, the cells were plated in six-well plates with 1 × 10^5^ cells and used when subconfluent. STC-1 cells were stimulated for 15 min with maleate/succinate (ranging from 10^−9^ M to 10^−5^ M) or a buffer control. The buffer was aspirated before and after 15 min of the stimulation period and analyzed for CCK content by bioassay.

### 4.4. Rat Pancreatic Acini for CCK Bioassay

Rat pancreatic acini were prepared by collagenase digestion. Pancreatic acini were incubated in 10 mM HEPES-buffered Ringer solution, containing 100 U/mL collagenase, 6 mM glucose, 0.5% bovine serum albumin, 0.1% soy bean trypsin inhibitor, and Eagles minimum amino acid supplements. The collagenase buffer was injected into the pancreatic interstitium to permit better digestion. After 30 min of incubation at 37 °C under O_2_ saturation in a shaking water bath, acini were dispersed with mild shearing forces, passed through double-filter gauze, purified by sedimentation, and resuspended in HEPES-buffered Ringer solution. CCK was quantified by a bioassay measuring amylase release after stimulation of isolated pancreatic acini [52,53]. For amylase release experiments, aliquots of the acini suspension were added to prepared vials containing the re-diluted eluates from the perfusions and cell culture experiments or CCK at known concentrations for construction of a standard curve. Vials were incubated in a shaking water bath at 37 °C under O_2_ for 10 min. The reaction was stopped by placing the vials on ice and separating the cells by centrifugation. Amylase release into the medium and total acinar–amylase content were measured by a colorimetric assay. Amylase release is expressed as a percentage of initial total amylase content. The amylase release from the unknown concentrations from the eluate of the perfused mucosal cells or supernatant from the STC-1 cells were taken into the standard curve constructed from the known concentrations of CCK, and thus, the bioactive CCK was determined in the unknown samples.

To confirm the amylase release from pancreatic acini, a specific CCK1 receptor-antagonist l-364718 (1µg/mL) [54] was added to the CCK bioassay, in combination with the eluates from mucosa cells or supernatants from STC-1 cells. The changes in intracellular calcium ion concentrations are receptor-regulated signals and precede secretions. To study the possible involvement of calcium channels in maleate and succinate-stimulated CCK release, we used diltiazem, which blocks l-type calcium channels [23]. STC-1 cells were incubated for 15 min with 10^−5^ M diltiazem before stimulation with succinate or maleate, and CCK was measured.

### 4.5. CCK Measurements by RIA

To exclude the possibility of amylase release by l-type calcium channel blocker diltiazem from pancreatic acini [55], we used an RIA kit for CCK measurements. STC-1 cells were stimulated for 15 min with maleate and succinate (2.5 mM) or a buffer control. The maximum CCK secretion was blocked by 10^−5^ M diltiazem hydrochloride. Supernatant was aspirated before and after 15 min of stimulation and analyzed for CCK by the RIA (RB 302 EURIA-CCK from Eurodiagnostic) kit as per the manufacturer’s instructions.

### 4.6. Lactate Dehydrogenase (LDH) Release from STC-1 Cells

To confirm physiological effects of maleate and succinate, cytotoxic effects of maleate and succinate were studied by measuring lactate dehydrogenase (LDH) release from STC-1 cells [3]. LDH is a cytosolic enzyme release into the culture medium upon cell death due to plasma membrane damage. Cells were cultured; 1 × 10^5^ cells/well were plated into six-well plates and used when subconfluent. Cells were washed twice with HEPES buffer and preincubated for 15 min with 1 mL of buffer at 37 °C, 5% CO_2_. The cells were treated with maleate or succinate and incubated for 15 min. After stimulation period, the cells were lysed with lysis buffer to obtain the maximum LDH activity and incubated for 45 min. Supernatants were collected for LDH measurements by a LDH cytotoxicity assay kit (Thermo Fisher Scientific, Waltham, MA, USA) as per user instructions, and the LDH release percentage was calculated according to the manufacturer’s instructions.

### 4.7. Statistical Analyses

Statistical analyses were done by one-way ANOVA for all the groups. Differences were considered statistically significant at *p* < 0.05 levels.

## 5. Conclusions

In conclusion, the present results identify succinate and maleate from fermented alcoholic beverages as physiological stimulants in CCK release, stimulating pancreatic exocrine secretions. We speculate that receptors on I-cells in the intestines detect succinate and maleate and that interference with or blockade of these receptors might be protective against pancreatic hypersecretion and pancreatitis.

## Figures and Tables

**Figure 1 ijms-21-00589-f001:**
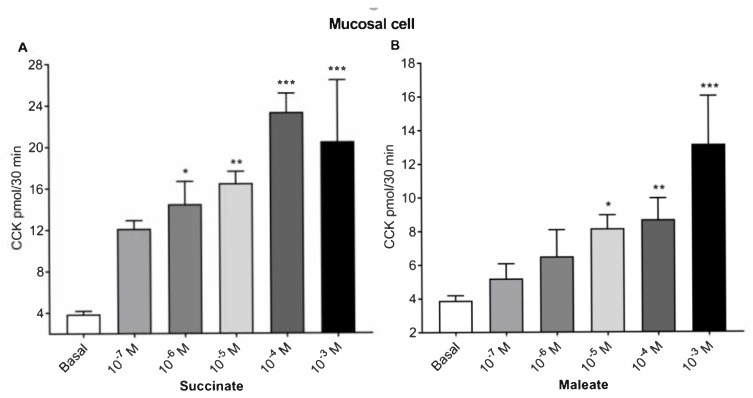
Dose responses of succinate and maleate in duodenal mucosal cells. Cholecystokinin (CCK) release from duodenal mucosal cells by succinate (**A**) and maleate (**B**). Succinate and maleate doses dependently stimulated CCK release from duodenal mucosal cells. CCK was measured in a bioassay and is expressed as a CCK release/30 min collection period. Results were expressed as mean ± SEM, *n* = 6. * *p* < 0.05, ** *p* < 0.01, *** *p* < 0.001, compared with basal values.

**Figure 2 ijms-21-00589-f002:**
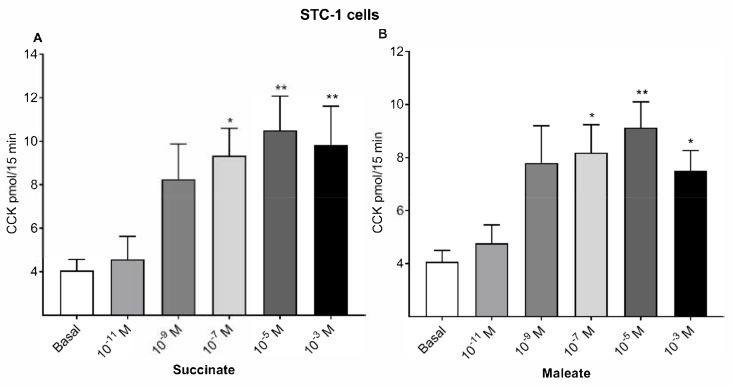
Dose responses of succinate and maleate in the enteroendocrine cell line STC-1. CCK release from STC-1 cells by succinate (**A**) and maleate (**B**). Succinate and maleate doses dependently stimulated CCK release from duodenal mucosal cells. CCK was measured in a bioassay and is expressed as a CCK release/15 min collection period. Results were expressed as mean ± SEM, *n* = 8. * *p* < 0.05, ** *p* < 0.01, compared with basal values.

**Figure 3 ijms-21-00589-f003:**
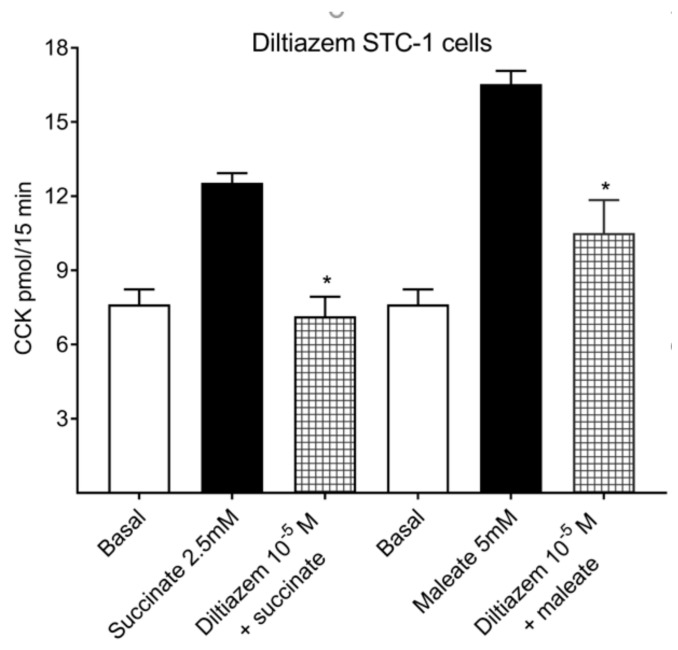
Effects of the L-type voltage-sensitive calcium channel blocker diltiazem on succinate and maleate in STC-1 cells by radioimmunoassay (RIA). Diltiazem 10^−5^ M reduced succinate-stimulated CCK secretions by 43% and maleate-induced CCK release by 36.4%, respectively. Results were expressed as mean ± SEM, *n* = 4. * *p* < 0.05.

**Figure 4 ijms-21-00589-f004:**
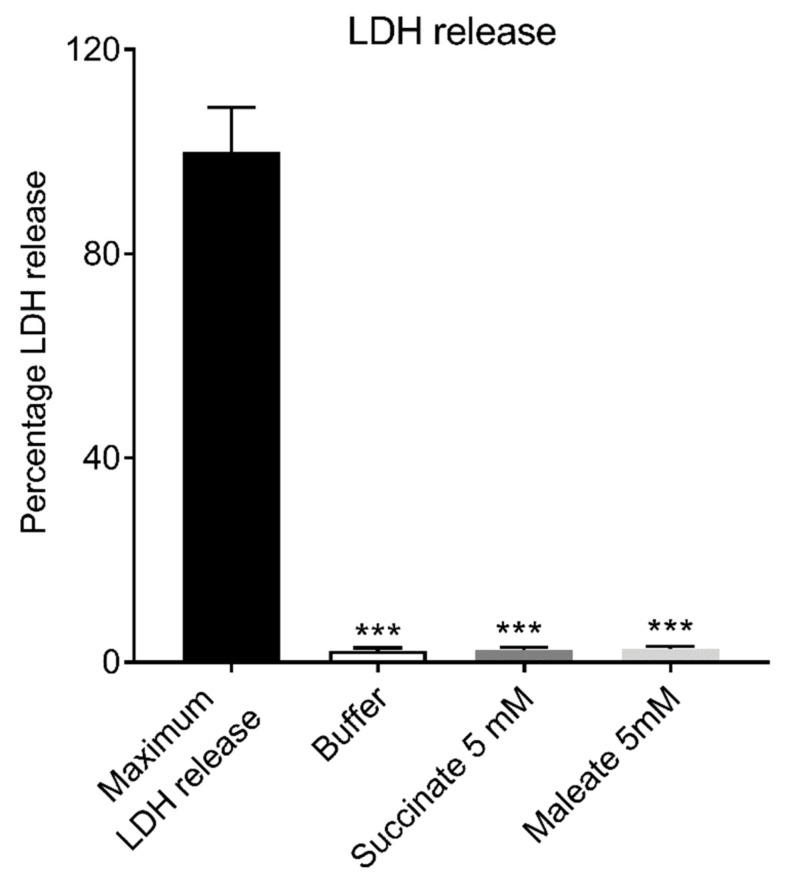
Both succinate and maleate inhibited lactate dehydrogenase (LDH) release from STC-1 cells. Results were expressed as mean ± SEM, *n* = 4. * *p* < 0.05, ** *p* < 0.01, *** *p* < 0.001.

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
