# Peer review of "Release of Cholecystokinin from Rat Intestinal Mucosal Cells and the Enteroendocrine Cell Line STC-1 in Response to Maleic and Succinic Acid, Fermentation Products of Alcoholic Beverages"

_ijms, 2020, doi:10.3390/ijms21020589_

Round 1

Reviewer 1 Report

The manuscript is of interest  because CCK is expressed in a cell-specific manner also outside the gastrointestinal tract and authors investigate the CCK releasing factors using  a validated experimental setting. However there are issues that need to be addressed:

Miao et al., reported that under physiological conditions the levels of malate and succinate are too low to activate the GPR91 [21]. Thus, in order to improve the quality of the manuscript should be tested if in the experimental setting succinate and malate activate GPR91 receptor and thereafter stimulates CCK release.   Moreover, to double check the specificity of the effects it should be tested if the CCK secretion is absent in calcium-free media.

Please provide the code number of the kits and the intra- and inter-assay CV. Where the samples analyzed in duplicates in the same assay to prevent inter assay variation?

Please provide figures 2, 3 and 5 that are missing in the manuscript.

Please update the references to more recent literature.

Please, to make the manuscript clear to the readers, it should be added throughout the manuscript  that L-364718 is a CCK1 receptor antagonist (Page 4 Line 6) . Same for Diltiazem. Moreover it should be specified the full name for LDH and that is a soluble cytosolic enzyme releases into culture medium upon cell death due to damage of plasma membrane.

Page 4 line 10 Pease clarify the sentence: Are the effects blocked by the  CCK1 receptor antagonist, L-364718 also antagonized in the pancreatic acini?

The discussion is poor and should be improved by more focusing on factors and mechanism of action that stimulates the CCK release.

Author Response

Miao et al., reported that under physiological conditions the levels of malate and succinate are too low to activate the GPR91 [21]. Thus, in order to improve the quality of the manuscript should be tested if in the experimental setting succinate and malate activate GPR91 receptor and thereafter stimulates CCK release.   Moreover, to double check the specificity of the effects it should be tested if the CCK secretion is absent in calcium-free media.

Response:

Thank you very much for kind suggestion.

GPR91 is widely expressed in intestine and succinate is endogenous ligand for the receptor (Regard et al. Cell 2008). During normal physiological conditions, the levels of succinate are below the reported EC50 of GPR91 (Husted et al. Cell Metab 2017); however, in metabolic stress conditions such as hyperglycermia and ischemia succinate level rises and enabling GPR91 activation (Sadagopan et al. Am J Hypert 2007; Chouchani et al. Nature 2014; Cummings et al. Am J Physiol 2014).

For the specificity, we have used the diltiazem calcium channel blocker, which inhibited the CCK secretion induced by succinate or maleate. We have previously shown that basal [C2+]i decreased in Ca2+-free bath solution, and under this condition the stimulus- e.g. GABA had no effect on [Ca2+]i (Glassmeier et al. J Physiol 1998). In addition, we have shown that the selectively GABAC receptors cis-4-aminocrotonic acid (CACA) stimulated [C2+]i, which was inhibited by the calcium channel blocker nifedipine (100 µM), and abolished in a Ca2+-free bath solution (Jansen et al. Eur J Physiol 2000). Furthermore, the group of Rozengurt demonstrated that the bitter stimuli denatonium benzoate (DB) is a potent stimulant of CCK release from enteroendocrine STC-1 cells and that treatment with either EGTA or nitrendipine prevented this effect, concluding that tastant-elicited CCK release is mediated by an increase in [Ca2+]i produced by the opening of L-type voltage sensitive calcium channels (Chen et al. Am J Physiol Cell Physiol 2006).

Regard J.B., Sato I.T., Coughlin S.R. Anatomical profiling of G protein-coupled receptor expression. Cell 2008, 135, 561–571.

Husted A.S., Trauelsen M., Rudenko O., Hjorth S.A., Schwartz T.W. GPCR-mediated signaling of metabolites. Cell Metabolism. 2017;25:777–796.

Sadagopan N., Li W., Roberds S.L., Major T., Preston G.M., Yu Y. Circulating succinate is elevated in rodent models of hypertension and metabolic disease. American Journal of Hypertension. 2007;20:1209–1215.

Chouchani E.T., Pell V.R., Gaude E., Aksentijevic D., Sundier S.Y., Robb E.L. Ischaemic accumulation of succinate controls reperfusion injury through mitochondrial ROS. Nature. 2014;515:431–435.

Cummins T.D., Holden C.R., Sansbury B.E., Gibb A.A., Shah J., Zafar N. Metabolic remodeling of white adipose tissue in obesity. American Journal of Physiology – Endocrinology And Metabolism. 2014;307:E262–E277.

Glassmeier G, Herzig KH, Höpfner M, Lemmer K, Jansen A, Scherubl H. Expression of functional GABAA receptors in cholecystokinin-secreting gut neuroendocrine murine STC-1 cells. J Physiol. 1998;510: 805-14.

Jansen A, Hoepfner M, Herzig KH, Riecken EO, Scherübl H. GABA(C) receptors in neuroendocrine gut cells: a new GABA-binding site in the gut. Pflugers Arch. 2000;441: 294-300.

Chen MC, Wu SV, Reeve JR Jr, Rozengurt E. Bitter stimuli induce Ca2+ signaling and CCK release in enteroendocrine STC-1 cells: role of L-type voltage-sensitive Ca2+ channels. Am J Physiol Cell Physiol. 2006;291(4):726-39.

Please provide the code number of the kits and the intra- and inter-assay CV. Where the samples analyzed in duplicates in the same assay to prevent inter assay variation?

Response:

The kits code number was RB 302 EURIA-CCK from Eurodiagnostic. The intra-assay CV was <14% and inter-assay CV 10% and within the limits provided by manufacturer.

Please provide figures 2, 3 and 5 that are missing in the manuscript.

Response:

We are very sorry but we are not sure why you did not see the figures, since all figures were in the manuscript and received by editorial office.

Please update the references to more recent literature.

Response:

We updated the literature were made to the manuscript as suggested.

Yamazaki, T., Morimoto-Kobayashi, Y., Koizumi, K., Takahashi, C., Nakajima, S., Kitao, S., Taniguchi, Y.; Katayama, M., Ogawa, Y. Secretion of a Gastrointestinal Hormone, Cholecystokinin, by Hop-Derived Bitter Components Activates Sympathetic Nerves in Brown Adipose Tissue. J. Nutr. Biochem. 2019;64:80-87.

Rehfeld, J.F. Cholecystokinin-from Local Gut Hormone to Ubiquitous Messenger. Front. Endocrinol (Lausanne) 2017;8:47.

Jeon, T.I., Seo, Y.K., Osborne, T.F. Gut Bitter Taste Receptor Signalling Induces ABCB1 through a Mechanism Involving CCK. Biochem. J. 2011;438:33-37.

Dufresne, M., Seva, C., Fourmy, D. Cholecystokinin and Gastrin Receptors. Physiol. Rev. 2006;86:805-847.

Williams, J.A. Receptor-Mediated Signal Transduction Pathways and the Regulation of Pancreatic Acinar Cell Function. Curr. Opin. Gastroenterol. 2008; 24:573-579.

Liang T, Dolai S, Xie L, Winter E, Orabi AI, Karimian N, Cosen-Binker LI, Huang YC, Thorn P, Cattral MS, Gaisano HY. Ex vivo human pancreatic slice preparations offer a valuable model for studying pancreatic exocrine biology. J Biol Chem. 2017;292:5957-5969.

Please, to make the manuscript clear to the readers, it should be added throughout the manuscript  that L-364718 is a CCK1 receptor antagonist (Page 4 Line 6) . Same for Diltiazem. Moreover it should be specified the full name for LDH and that is a soluble cytosolic enzyme releases into culture medium upon cell death due to damage of plasma membrane.

Response:

Thank you! We changed our manuscript accordingly. Highlighted on page 4, 5 and 6.

Page 4 line 10 Please clarify the sentence: Are the effects blocked by the  CCK1 receptor antagonist, L-364718 also antagonized in the pancreatic acini?

Response:

The effects of CCK release was antagonized by the CCK1 receptor antagonist L-364718. This has now been changed in the manuscript page 4.

The discussion is poor and should be improved by more focusing on factors and mechanism of action that stimulates the CCK release.

Response:

We now changed the manuscript and added new sections on page 6.

Reviewer 2 Report

The authors demonstrate maleic acid and succinic acid release CCK from rat intestinal mucosal cells and STC-1 cells. This provide useful information for alcoholic abuse treatment. There are many problems in the current contents.

Major points

1. CCK1R-bearing cell does not release CCK. Why was CCK release decreased from intestinal mucosal cells and STC-1 cells by CCK1R antagonist? The authors should mention this point.

2. Does the CCK released from intestinal mucosal cells or STC-1 cells affect pancreatic enzyme secretion from pancreatic acinal cells? The authors should show direct experiments.

3. The authors did not investigate relation between CCK and PKD1. The authors did not investigate relation between maleic acid or succinic acid and other GI hormones, such as histamine, gastrin etc. So, these discussions are not need in the Discussion part.

Author Response

The authors demonstrate maleic acid and succinic acid release CCK from rat intestinal mucosal cells and STC-1 cells. This provide useful information for alcoholic abuse treatment. There are many problems in the current contents.

Major points

1. CCK1R-bearing cell does not release CCK. Why was CCK release decreased from intestinal mucosal cells and STC-1 cells by CCK1R antagonist? The authors should mention this point.

Response:

Intestinal STC-1cells have been an established model for studying various compounds and mechanisms for cholecystokinin secretion(Chang  et al. 1994). Takata et al., showed that STC-1 cells expressed CCK-1 receptor as well as its peptide-ligand, CCK (Takata et al. J Gastroenterol 2002).  Geraedts et al., reported that sucralose and other sweeteners increase CCK and GLP-1 secretion in STC-1 cells (Geraedts et al. Mol Nutr Food Res 2012). In both in vitro and in vivo studies, secretion of various isoforms of CCK from enteroendocrine cells has been reported  Yoon et al. Endocrine 1994 ) Yoon et al. Endocrinology 1997)  differing in activity at CCK-1 receptor. The pancreatic secretion in rodents is mediated by direct activation of CCK-1 receptors located on acinar cells as well as on vagal afferents (Li et al. J Clin Invest  1993; Singer et al. Cell Biol Int  2009 and Wang et al. Am J Physiol Regul Integr Comp Physiol  2007).

Chang CH., Chey W Y., Sun Q., Leiter A., Chang TM. Characterization of the release of cholecystokinin from a murine neuroendocrine tumor cell line, STC-1. Biochim. Biophys. Acta. 1994;1221:339–347.

Geraedts M., Troost F., Saris W. Addition of sucralose enhances the release of satiety hormones in combination with pea protein. Mol Nutr Food Res 2012; 56:417–424.

Takata Y, Takeda S, Kawanami T, Takiguchi S, Yoshida Y, Miyasaka K, Funakoshi A. Promoter analysis of human cholecystokinin type-A receptor gene. J Gastroenterol. 2002;37(10):815-20.

Yoon JY., Beinfeld, MC. A murine intestinal endocrine tumorcell line, Stc-1, expresses Cck, Pc1 and Pc2 messenger-RNA, processes Pro-Cck to Cck-8 and displays camp-regulated release. Endocrine. 1994;2:973–977.

Yoon, J., Beinfeld, M. C. Prohormone convertase 2 is necessary for the formation of cholecystokinin-22, but not cholecystokinin8, in RIN5F and STC-1 cells. Endocrinology. 1997;138:3620–3623.

Li Y., and Owyang C. Vagal afferent pathway mediates physiological action of cholecystokinin on pancreatic enzyme secretion. J Clin Invest. 1993;92:418-424.

Singer MV., and Niebergall-Roth E. Secretion from acinar cells of the exocrine pancreas: Role of enteropancreatic reflexes and cholecystokinin. Cell Biol Int. 2009;33:1-9.

Wang BJ, and Cui ZJ. How does cholecystokinin stimulate exocrine pancreatic secretion? From birds, rodents, to humans. Am J Physiol Regul Integr Comp Physiol. 2007; 292:666-678.

2. Does the CCK released from intestinal mucosal cells or STC-1 cells affect pancreatic enzyme secretion from pancreatic acinal cells? The authors should show direct experiments.

Response:

It is well known that CCK stimulates pancreatic enzyme secretion as the major stimulus of the intestinal phase of pancreatic secretion. It has been shown that CCK-1 receptor is present with distinct cellular distributions in the exocrine pancreas of rodents, guinea-pigs and humans (Jensen et al. Proc Natl Acad Sci USA 1980; Galindo et al.  Pancreas 2005; Liang  et al. J Biol Chem. 2017). Physiologic concentrations of CCK, elicit oscillatory increases in cytosolic calcium and stimulation of enzyme secretion on human pancreatic acinar cells (Murphy et al. Gastroenterology 2008).

We have directly demonstrated that CCK released from intestinal mucosal cells stiumulated amylase secretion in our bioassay system.

We agree that the expression of the various receptor types is different in different species. In humans Liang et al showed that CCK8 stimulated amylase secretion which was inhibited by the CCK-1 receptor antagonist (devazepide) but not by the CCK-2 receptor antagonist (L365,260) (Liang et al. J Biol Chem 2017).

Jensen RT., Lemp GF., Gardner JD. Interaction of cholecystokinin with specific membrane receptors on pancreatic acinar cells. Proc Natl Acad Sci USA. 1980;77:2079–2083.

Galindo J., Jones N., Powell GL., Hollingsworth SJ., Shankley N. Advanced qRT-PCR technology allows detection of the cholecystokinin 1 receptor (CCK1R) expression in human pancreas. Pancreas. 2005;31:325–331.

Liang T., Dolai S., Xie L., Winter E., Orabi AI., Karimian N., Cosen-Binker LI., Huang YC., Thorn P., Cattral MS., Gaisano HY. Ex vivo human pancreatic slice preparations offer a valuable model for studying pancreatic exocrine biology. J Biol Chem. 2017;292(14):5957-5969. 

Murphy JA., Criddle DN., Sherwood M., Chvanov M., Mukherjee R., McLaughlin E. et al. Direct activation of cytosolic Ca2+ signaling and enzyme secretion by cholecystokinin in human pancreatic acinar cells. Gastroenterology. 2008;135:632–641.

3. The authors did not investigate relation between CCK and PKD1. The authors did not investigate relation between maleic acid or succinic acid and other GI hormones, such as histamine, gastrin etc. So, these discussions are not need in the Discussion part.

Response:

The discussion is modified as suggested accordingly.

Round 2

Reviewer 2 Report

previous concern 1:

The authors do not mention why CCK1R antagonist inhibits CCK release from duodenal mucosal cells or STC-1 cells. Does CCK promote CCK release from CCK-1R expressing cells?

previous concern 2:

The authors do not show that the secreted CCK really promoted amylase release. For example, did the condition medium from succinate stimulating STC1 culture promote amylase release from pancreatic acinal cells?

Author Response

Previous concern 1

The authors do not mention why CCK1R antagonist inhibits CCK release from duodenal mucosal cells or STC-1 cells. Does CCK promote CCK release from CCK-1R expressing cells?

Response:

We apologize for not clearly answering the questions before.

The CCK 1R antagonist was used to make sure that we indeed measure only interaction with the CCK 1 R since we have been using a bioassay blocking CCK1R on the isolated acini. Please see chapter 4.4. last paragraph: To confirm the amylase release from pancreatic acini, a specific CCK1 receptor antagonist L-364718 (1µg/ml) [58] was added to the CCK bioassay in combination with the eluates from mucosa cells or supernatants from STC-1 cells.

Previous concern 2:

The authors do not show that the secreted CCK really promoted amylase release. For example, did the condition medium from succinate stimulating STC1 culture promote amylase release from pancreatic acinal cells?

Response:

In our response to your second comment, we wrote that

We have directly demonstrated that CCK released from intestinal mucosal cells stimulated amylase secretion in our bioassay system.

In chapter 2.2. last sentence we wrote that succinate did not affect amylase release from the pancreatic acinar cells. “In addition, no direct effect on amylase secretion was observed with maleate or succinate on freshly isolated pancreatic acini”

Round 3

Reviewer 2 Report

concern 1:

I agree with the authors’ comments.

concern 2:

The authors wrote “The CCK1 receptor antagonist (L-364718) completely inhibited amylase release from pancreatic acini stimulated by the eluates from intestinal mucosal cells (Figure 3A) or from STC-1 cells (Figure 3B)”. However, the vertical axis of Fig 3 is not amylase release but CCK release. The authors did not show the amylase release data.

Author Response

Concern 2:

The authors wrote “The CCK1 receptor antagonist (L-364718) completely inhibited amylase release from pancreatic acini stimulated by the eluates from intestinal mucosal cells (Figure 3A) or from STC-1 cells (Figure 3B)”. However, the vertical axis of Fig 3 is not amylase release but CCK release. The authors did not show the amylase release data.

Response:

We agree with the referee that sentence 2.2. CCK release by succinate and maleate is antagonised by the CCK1 receptor antagonist is indeed confusing and should be omitted. We therefore now clarified this in the manuscript accordingly and omitted figure 3.

In now reads in the manuscript on page 4:

Additional factors released from the intestinal mucosa and STC-1 cells may also influence amylase release from pancreatic acini. The CCK1 receptor antagonist (L-364718) completely inhibited amylase release from pancreatic acini in the bioassay from the eluates from intestinal mucosal cells and the supernatant from STC-1 cells; therefore, CCK values did not differ from the basal. In addition, no direct effect on amylase secretion was observed with maleate or succinate on freshly isolated pancreatic acini.

Furthermore, we added an additional sentence to the description of bioassay on page 7.

In 4.4. Rat pancreatic acini for CCK bioassay – we described the bioassay:

CCK was quantified by a bioassay measuring amylase release after stimulation of isolated pancreatic acini [56,57]. For amylase release experiments, aliquots of the acini suspension were added to prepared vials containing the re-diluted eluates from the perfusion and cell culture experiments or CCK at known concentrations for construction of a standard curve. Vials were incubated in a shaking water bath at 37°C under O2 for 10 min. The reaction was stopped by placing the vials on ice and separating the cells by centrifugation. Amylase release into the medium and total acinar amylase content was measured by a colorimetric assay. Amylase release is expressed as percentage of initial total amylase content. The amylase release from the unknown concentrations from the eluate of the perfused mucosal cells or supernatant from the STC-1 cells were taken into the standard curve constructed of the known concentration of CCK and thus the bioactive CCK was determined in the unknown samples.